# A Comparative Analysis of Machine Learning Techniques for Muon Count in UHECR Extensive Air-Showers

**DOI:** 10.3390/e22111216

**Published:** 2020-10-26

**Authors:** Alberto Guillén, José Martínez, Juan Miguel Carceller, Luis Javier Herrera

**Affiliations:** 1Computer Technology and Architecture, University of Granada, 18071 Granada, Spain; jherrera@ugr.es; 2Cosmos and Theoretical Physics Department, Univerisity of Granada, 18071 Granada, Spain; jcarlosmv@correo.ugr.es (J.M.); jmcarcell@ugr.es (J.M.C.)

**Keywords:** machine learning, Pierre Auger Observatory, muon count, regression, LSSVM

## Abstract

The main goal of this work is to adapt a Physics problem to the Machine Learning (ML) domain and to compare several techniques to solve it. The problem consists of how to perform muon count from the signal registered by particle detectors which record a mix of electromagnetic and muonic signals. Finding a good solution could be a building block on future experiments. After proposing an approach to solve the problem, the experiments show a performance comparison of some popular ML models using two different hadronic models for the test data. The results show that the problem is suitable to be solved using ML as well as how critical the feature selection stage is regarding precision and model complexity.

## 1. Introduction

The way in which ultra-high-energy cosmic rays (UHECRs) are originated is one of the main mysteries nowadays in Astroparticle Physics. To understand these particles, the Pierre Auger Observatory [1] was designed and built. A very ambitious project was started that ended up as the biggest experiment in the world. A large area of 3000 square kilometers is instrumented with water-Cherenkov Detectors (WCDs) that are able to record the signals generated by the particles that reach the ground as they travel through the water of the WCDs.

The interactions of UHECRs with molecules of air at the atmosphere produce what is known as Extensive Air Showers (EAS): the primary particle collides at the top of the atmosphere and generates a cascade of secondary particles like photons, electrons, positrons and muons.

From the signals measured, scientists have to figure out answers to many questions, such as: what type of particle arrived at the top of the atmosphere and where it came from. To answer the first question, it is key to know how many muons were generated during the shower development in the atmosphere. As particles collide in the atmosphere and reach the ground, they generate on each WCD a signal which is a combination of the signal generated by the electromagnetic component of the shower and the muonic one. Thus, if it is desired to estimate the particle nature using the muonic component, it becomes a challenge with the devices available nowadays.

This paper deals with the problem of using machine learning techniques to analyze the signals in WCDs and compute the contribution of muons to the total recorded signal. After describing how the problem has been tackled and the data used in Section 2, all the models used are presented in Section 3. Finally, experiments are shown in Section 4 and Conclusions are drawn.

## 2. Problem Definition and Data Description

This work is based on the use of simulated data. Figure 1 summarizes all the stages of the simulation process. The package in control of how the extensive-air shower develops in the atmosphere is CORSIKA [2]. Hadronic interactions are modelled through the use of QGSJET-II [3] or EPOS-LHC [4]. The signals left in the detectors by the particles that traverse them are generated using the Auger $\bar{\mathrm{Off}}$$\underline{\mathrm{line}}$ software [5]. Finally, the ROOT package [6] is used to store the data that will be used as input for the machine learning methods.

Each simulation requires high computational time and a large amount of disk space, thus, a reduced amount of data is available. Nonetheless, the simulations computed can be enough for the models used. It is worth noting that these simulators include noise to make the simulation closer to reality. Furthermore, $\bar{\mathrm{Off}}$$\underline{\mathrm{line}}$ models all the electronic components (such as triggers, sensors, and other analogue elements included in the real detector) so few aspects from the real world are left behind. Regarding the simulations available done with the QGSJET-II package, there are four types of primary nuclei: for proton, oxygen, helium and iron, there are over 20,000 samples. This number was divided into two subsets that will be used respectively for training and testing. The final number of samples for each primary is:Helium, training: 16,007, test: 4001. Total: 20,008.Iron, training: 16,019, test: 4004. Total: 20,023.Oxygen, training: 16,021, test: 4005. Total: 20,026.Proton, training: 16,026, test: 4006. Total: 20,032.

The algorithms will be trained and tested using these data. For the tests, however, another testing stage will be carried out using the data available from simulations using the EPOS LHC model. In this way, it is possible to test if the models are able to generalise properly the natural phenomenon modelled independently of the simulator used to generate the data. Regarding the data available in this case, there are 86,923 showers initiated by iron nuclei and 78,659 by protons.

### 2.1. Problem Definition

As described above, there are some inputs that consist of the simulation of a particle being recorded by the WCD. That signal encompasses the muonic component μ as well as the electromagnetic component em. The question is how to obtain the muonic component to determine the composition of the primary particle. As stated, it seems straightforward as a blind source separation problem which can be tackled using Independent Component Analysis. The problem can be thought as a room with several microphones, each one recording in a different position. The room is the tank and each microphone is one of the three PMTs inside one tank. From this point of view, the direct application of ICA seems straightforward (although there are some differences as the signals in water behave differently than in air). After some trials, we could not find a successful solution as the standard ICA (using implementation of the FastICA [7] algorithm provided by Sci-kit learn toolbox [8]) algorithm was not able to separate the muonic and the electromagnetic component. Therefore, the proposed solution was to map this problem to a ML classical approach. In this case, the muonic signal can be integrated, providing a continuous value. Thus, it is possible to end up with a set of inputs ${\vec{x}}_{i}\in R^{d},i=1,\ldots ,m$ (particles of the EAS interacting in a surface detector) and a continuous output value $Y=[y_{i}],i=1,\ldots ,m$ which corresponds to the integral of the muonic signal generated.

With this formulation, the mapping from Physics to ML results in a classical regression problem where it is desired to obtain a function *f* such that $f\left(\vec{x_{i}}\right)\approx y_{i},\forall \;i$.

#### Feature Extraction

As working with the raw signal might be too expensive in terms of memory and CPU requirements, the variables used to define ${\vec{x}}_{i}$ where taken from the output of the $\bar{\mathrm{Off}}$$\underline{\mathrm{line}}$ software package [5]. Nevertheless, according to the experts, the trace can be fully characterized by adding some features that can help the model to learn the function *f*. For example, the Risetime is used to infer information about mass composition in [9,10]. In these examples, the 〈Δ〉 method is presented performing a feature engineering. The Risetime has been used in later studies [11]. Thus the subset of variables remained as follows:

**Input variables**:Monte Carlo Energy *E*: the total energy (in EeV, Exaelectron Volts) of the primary cosmic ray. It has been transformed applying $\log_{10}$.Monte Carlo Zenith angle: angle in degrees between the zenith and the trajectory of the primary cosmic ray.Distance to the core *r*: distance between each station and the reconstucted position of the core of the shower at the ground, given in meters.Total signal $S_{total}$: real value in Vertical Equivalent Muons (VEMs) of the signal retrieved by the WCD.Trace length: number of bins with signal recorded (each bin considers 25 nanoseconds).

**Engineered variables**:Azimuth Angle ζ: measured in radians.Signal Risetime $t_{1/2}$: measured in nanoseconds [9].Signal Falltime: real value showing when the signal starts falling.Area over peak of the signal: sum of the signals in each trace divided by the maximum value of each trace.

**Output**:Muonic signal: in VEMs.

### 2.2. Normalization

Each row, corresponding to an event, has several columns that contain different magnitudes, as detailed above. Therefore to avoid problems with distances and error measures, the data has been normalised using the Z-Score (mean 0 and std 1).

One of the main advantages of this method, in comparison with Min-Max, is that test values that might be out of the range of training or validation will not get unrealistic representations.

## 3. Models Considered and Design Issues

This section will describe briefly the models that have been used in the comparison, and the criteria used to set the hyperparameters. For each model, two sets of hyperparameters were chosen depending on the subset of variables selected as described in detail in the next section.

### 3.1. Linear Regression (Lr)

The model known as LR is defined as:(1)$$y_{i}=\beta_{0}+\sum_{j=1}^{d}\beta_{j}x_{i}^{\left(j\right)}+\varepsilon$$
where $\beta_{i}$ are the coefficients that weigh each regressor, $\beta_{0}$ the intercept and ε the noise. The idea is to find a hyperplane that fits linearly the target output.

Due to the complexity of this natural phenomena, it is quite probable that non-linear relationships arise in the problem. However, it is interesting to maintain the linear approach due to its simplicity and interpretability.

The implementation used for this model is the one available at the SciKit Learn library [8].

### 3.2. Decision Tree (DT)

These types of trees are built based on partitions of the input space. The more partitions, higher accuracy can be obtained. There are several algorithms that divide the input space such as CART (Classification And Regression Trees) [12], C4.5 [13], CHAID [14], etc.

One of the main advantages of these models is the interpretability coming from simple if/else rules associated with the created partitions. However, this approach, in general, provides a lower accuracy in comparison with other methods.

Regarding the design of these models, the hyperparameter considered for computing the results shown in Section 4 is the maximum depth for the tree. For this parameter, all values in the range [1,50] were tested. The values chosen for the comparison were 12 for the first variable selection and 14 for the second.

### 3.3. Random Forest (RF)

The idea of this model is to build several decision trees to overcome their limited accuracy and combine them in an ensemble. Each subtree uses a different subset of regressors so, even though a subtree has less information than a DT that uses all the regressors, the combination of them outperforms the original approach.

However, this improvement in performance has the inconvenient of losing interpretability. As the number of subtrees increases, it is much more challenging to understand the final output of the model.

The hyperparameters considered to obtain the results were the maximum depth the forest can have and the number of estimators. The values considered for these parameters were ∈[1,50] for the depth and ∈[50,550] with a step of 50. The best value obtained was 500 for the number of estimators and 20 for the first variable selection and 450 estimators with 25 of maximum depth for the second variable selection.

### 3.4. Support Vector Regressor (SVR)

The model known as SVR is based on the Support Vector Machine for classification. The latter tries to find the support vector that linearly separates the input data. However, it is not always possible to do it, so these models use the kernel trick [15], projecting the input space into a higher dimensional feature space where the linear separation might be almost achieved. This ‘almost’ is regulated by parameters that regulate how many misclassifications can be made.

In the regression case, the parameter that regulates the SVR determines the maximum error that can be made when approximating an input vector.

Thus, the SVR proposed by Vapnik in [16] can be looked at a generalization of the original approach but providing a continuous output instead a label within a given subset.

The generalisation is achieved by introducing the ϵ that, together with the hyperplane, defines a tube that fits the output while keeping a permited error.

Formally, the problem consists of:(2)$$\min_{w,b,\varepsilon ,\varepsilon *}\frac{1}{2}w^{T}w+C\sum_{i=1}^{m}\varepsilon_{i}+C\sum_{i=1}^{m}\varepsilon_{i}^{*}$$

subject to the following constraints:(3)$$\begin{array}{c} w^{T}\phi \left({\vec{x}}_{i}\right)+b-y_{i}\leq \varepsilon +\varepsilon_{i}^{*}, \end{array}$$(4)$$\begin{array}{c} y_{i}-w^{T}\phi \left({\vec{x}}_{i}\right)-b\leq \varepsilon +\varepsilon_{i}^{*}, \end{array}$$(5)$$\begin{array}{c} \varepsilon_{i},\varepsilon_{i}^{*}\geq 0,\;i=1,\ldots ,m. \end{array}$$
where ϕ is the kernel function and *c* and ϵ the parameters that control how much flexibility to errors the model should have.

This paper has used the implementation available at [8] which integrates the libsvm presented in [17].

For further details, please refer to [18] and [19]. The number of parameters considered to evaluate SVRs were:Type of kernel function: Radial Base, Polynomial (from 3 to 5 degrees) and Sigmoid.*C*: values ∈ [10,100] in steps of 10.ϵ: values ∈[1e−4,1e−3,1e−2,1e−1].

The best values obtained to carry out the comparisons were:Type of kernel function: Radial Basis Function (Gaussian)*C*: 60ϵ: 0.1
for the first variable selection and:Type of kernel function: RBF*C*: 10ϵ: 0.1
for the second.

### 3.5. eXtreme Gradient Boost (XGBoost)

Gradient Boosted Regression Trees (GBRTs), also known as Gradient Boosting Machines (GBMs), were first presented in [20]. Among the different approaches for boosted trees available, the one presented in [21] and available at [22] provides effective results with an optimised implementation.

This type of model creates an ensemble of decision trees which is built in an incremental way like greedy algorithms. In the first stage, a weak tree that makes a poor approximation (always returning the average value) is formed. Afterwards, from the errors obtained a gradient is computed, and a new tree learns that gradient, and so on.

Let L(θ) be a loss function and obj(θ) be the objective function to be minimised by the model, which includes a regularisation term Ω(θ):(6)obj(θ)=L(θ)+Ω(θ)

The regularisation term allows to avoid overfitting which usually occurs when performing Gradient Boosting [20]. The output provided by the ensemble of trees if given as:(7)$$\hat{y}=\sum_{k=1}^{K}f_{k}\left(\vec{x_{i}}\right)$$
where $f_{k}$ is the output of a single tree (which has independent structure *q* and weights *w* for the leaves). As these parameters cannot be optimised in Euclidian space, one option is to follow a greedy iterative approach to minimise:(8)$$L^{\left(t\right)}=\sum_{i=1}^{n}l(y_{i},{\hat{y}}^{\left(t+1\right)}+f_{t}\left(\vec{x_{i}}\right)+\Omega \left(f_{t}\right)$$
where *t* refers to the iteration number. By reformulation the previous equation, it is possible to obtain a score which indicates the impurity of a tree and compare it with other trees to make a selection.

Thus, there is a wide variety of hiperparemeters to be optimised and some values regarding the structure of the tree and the ensemble (maximum tree depth, number of estimators), their values (learning rate for the leave weights) and how they split (minimum impurity decrease), among others.

To carry out the experiments to compare with the other models, the parameter selection was:maximum tree depth: values ∈[5,55] in steps of 5learning rate: values [0.001, 0.01, 0.1]number of estimators: values ∈[50,550] in steps of 5minimum impurity decrease: $0.05*std(y_{train})$
and after performing the training stage, the best configuration achieved was:maximum tree depth: 45learning rate: 0.01number of estimators: 500minimum impurity decrease: 10
for the first variable selection and:maximum tree depth: 20learning rate: 0.1number of estimators: 50minimum impurity decrease: 17
for the second.

### 3.6. Single and Multi Layer Perceptron

Artificial Neural Networks (ANNs) have been used widely in classification and regression tasks.

These models are inspired in biological neural networks, representing a mathematical simplification of those. The first approach consisted on the modeling of an artificial neuron and corresponded to McCulloh and Pitts in [23]. The output of such neuron can be seen as a weigthed sum of the activation function when it receives an input, formally:(9)$$\hat{y_{i}}=\phi \left(\sum_{j=1}^{d}w_{j}x_{i}^{\left(j\right)}+w_{0}\right).$$

Instead of using just one neuron, or unit, several can compute their corresponding activation functions grouped in the same layer (like Radial Basis Function Neural Networks [24]) or can have more layers (hidden layers) that can be trained using back-propagation like in [25]. Hence, the *j*-th unit in the layer h+1 receives as input:(10)$${x^{\prime }}_{j}^{h+1}=\sum_{k=1}^{U_{h}}{y^{\prime }}_{k}^{h}w_{ji}^{h}$$
where $w_{ji}$ is the corresponding weight for those two units and $y^{\prime h}=\phi^{h}\left(x^{\prime h-1}\right)$ the output of the units of the previous layer.

There have been recent advances thanks to the use of GPUs and stochastic training [26]. Nonetheless, a single layer still remains as a feasible architecture (Extreme Learning Machines are based on this architecture [27]). Therefore both shallow and deep configurations have been considered for this problem.

Other aspects that were evaluated as critical aspects for the design of the network are the number of units and the learning rate used to update the weights during the parameter adjustment. This algorithm iterates until it reaches convergence or the maximum number of iterations (epochs). For this work, the implementation of a MLP available (MLPRegressor) from [8] has been used with the following configurations:Activation function: ReLuArchitectures evaluated: [(10, 10, 10, 10), (50, 50, 50), (100, 100), (250)]Learning rate: Adaptive using α∈{0.0001,0.001,0.01}Max. epochs: 300
and after performing the training stage, the best configuration achieved was:Architecture: (50, 50, 50)Learning rate: α=0.01
for the first variable selection and:Architecture: (250)Learning rate: α=0.001
for the second method.

## 4. Experiments and Discussion

This section first describes how the dimensionality of the problem was reduced and then, it shows a comparison of the models previously described based on tests.

### 4.1. Feature Selection

As there are new variables proposed by the experts and due to the curse of dimensionality, it is desirable to reduce the number of them. For that purpose, an analysis of the linear correlation between the variables was done.

As Figure 2 shows, the maximum correlation between the output variable and a regressor is achieved by the Total Signal. The maximum correlation between regressors is of 0.77 obtained by the energy and the distance to the core. However, this value, although being high, is considered lower than a threshold decided in accordance with the experts already cited (α=0.85). To complement this analysis, two additional methods were tested in the search for dimensionality reduction for this problem.

#### 4.1.1. Using Mutual Information

The idea of using Mutual Information to carry out feature selection has been proposed in the literature in several ways [28,29,30,31]. Let $X=\{{\vec{x}}_{k}\}$ and $Y=\{y_{k}\}$ for k=1…n be the input sample pairs, then, the Mutual Information between these two variables can be defined as the amount of information that can be extracted from *Y* given *X*. It is formally defined as:(11)I(X,Y)=H(Y)−H(Y|X)=H(X)+H(Y)−H(X|Y).
where H() is the entropy. In the case of dealing with continuous variables, this entropy is defined as:(12)$$H\left(Y\right)=-\int \mu_{Y}\left(y\right)\log\mu_{Y}\left(y\right)dy,$$
(13)$$H\left(X|Y\right)=-\int \mu_{X}\left(x\right)\int \mu_{Y}\left(y|X=x\right)\log\mu_{Y}\left(y|X=x\right)dydx.$$
where $\mu_{Y}\left(y\right)$ is the density marginal function, leading to:(14)$$I\left(X,Y\right)=\int \mu_{X,Y}\left(x,y\right)\log\frac{\mu_{X,Y}\left(x,y\right)}{\mu_{X}\left(x\right)\mu_{Y}\left(y\right)}dxdy.$$

Hence, to be able to obtain a value for the MI, it is required to estimate the the joint probability density function between *X* and *Y*. Although there are several procedures to do so [32], one of the most popular is the one based in the *K* nearest neighbours proposed by [33].

A well-known approach to use MI to perform variable selection is the minimum redundancy and maximum relevance one (mRMR) [34].

The algorithm is based on the following heuristic: minimise redundancy among features but maximise relevance. Thus, it is possible to crop features already represented by others but keeping the features that are most important to the output. The criterion to determine both metrics is based on the mutual information defining redundancy as:(15)$$R=\frac{1}{{\left|S\right|}^{2}}\sum_{x_{i},x_{j}\in S}I\left(x_{i},x_{j}\right)$$
and relevance as:(16)$$D=\frac{1}{\left|S\right|}\sum_{x_{i}}I\left(x_{i},Y\right)$$
where *S* is the subset of features being evaluated.

This approach has several advantages like the feasibility of finding a solution faster than evolutive approaches and that mRMR can be considered as an optimal first-order approximation of the mutual information.

Table 1 shows the feature ranking using the mRMR algorithm where the higher the score is, the better is the variable to approximate the output.

The first six variables ranked have similar score meanwhile the rest decreases significantly.

#### 4.1.2. Using XGBoost

Table 2 shows the importance of each variable according to XGBoost. This importance is given by the number of times each variable is used in all the subtrees built. The higher the importance is, the more relevant the feature.

After comparing the results, it seems reasonable to keep the first six variables in both methods and compare them, thus the following variables are kept (the experiments carried out with this subset of variables are notated as VS2):Monte Carlo Energy *E*.Monte Carlo Zenith angleDistance to the core *r*.Total signal $S_{total}$.Signal Risetime $t_{1/2}$.Area over the Peak.

### 4.2. Model Comparisons

After obtaining the data from the two hadronic models (EPOS LHC and QGSJET-II), some of them were used for training and some others were kept to be used in the test comparison. The reason to consider only QGSJET-II for training is that there are more types of primary particles available for training in comparison with EPOS LHC (only proton and iron nuclei).

After performing the feature selection considering the two criteria described previously (mRMR and XGBoost), the training data was used to select the best hyperparameters that each model requires. Once the hyperparameters were fixed, as detailed in Section 3, a 100-fold cross-validation procedure was carried out for each type of model considered to obtain mean performances according to several criteria: Mean Absolute Error (MAE) $\left|y-\hat{y}\right|$, Mean Squared Error (MSE) and $R^{2}$ (determination coefficient). Table 3 and Table 4 show the results for the mean and standard deviations obtained when looking only to the validation data during the 100-fold Cross-Validation (CV).

The work flow followed to obtain the final result is depicted in Figure 3.

Table 5 shows the space in permanent storage after serializing the corresponding model objects using the method dump from the joblib library [35].

#### 4.2.1. ANOVA Results for MAE, MSE and $R^{2}$

In order to determine which model/algorithm performed the best, a test ANOVA [36] was carried out to see if there were significant differences in the results. The analysis was based on a one-way ANOVA considering as unique factor the model used and the dependent variable, the approximation error (considering isolatedly MAE, MSE and $R^{2}$). The null hypothesis established is that both models behave equally well and the differences in the errors are due to randomness introduced by hyperparameters initialization.

Before applying ANOVA, a test (Shapiro–Wilk [37]) to check if the distribution of the results belonged to a normal distribution (as ANOVA requires) was applied. Results showed *p*-values over 0.05 so it is feasible to assume that the error measures in the CV fall within a normal distribution.

Figure 4, Figure 5 and Figure 6 show the ANOVA p-values for the results obtained after the 100-fold CV using the two types of feature selection. The less informative criterion is $R^{2}$ as it shows very few differences in comparison with the other two. Dark blue means that there is a statistical difference of one model versus the other (*p*-value < 0.05).

#### 4.2.2. Discussion

From the statistical point of view, after considering the ANOVA tables, it is not possible to claim a clear winner. However, it is possible to identify the models that perform worse. For this non-linear problem, LR has a poor performance in comparison with the other approaches and DT are always outperformed by its improvements like RF and Boosted trees.

Taking a closer look to Table 3 and Table 4, it is possible to find a minimum for the MSE criterion. The best result is provided by the SVR and, in second place, MLP, both showing as well the smallest values for the standard deviation. In consequence, for the next section (the test comparison of the two hadronic models), only the results provided by the SVR will be used.

It is interesting that, even in the ANOVA tables, the variable selection procedure is important and just by swapping one variable can lead to better results. In this case, the variable selection provided by the XGBoost allows all the models to obtain better results. This is remarkable and should be a warning for those practitioners considering only the mutual information as a valid criterion, when the reality is that the values used are just an estimation of it.

A usually ignored aspect about the models is the complexity or implementation restrictions that they might present. Table 5 shows significant differences in the memory requirements for the different models. The more data-driven is the model, the higher the space it requires. In the case of implementing an ad-hoc system to detect anomalies in real measurement devices like the ones being installed in the Pierre Auger Observatory (AugerPrime [38]), this element should be considered carefully. From this point of view, there is no doubt regarding the model that should be used: Neural networks. The MLP used in this paper without considering a reduced subset of variables fits in a few Kibibytes. Nonetheless, considering today’s memory capacities and seeing the good performance achieved by SVR, it is advisable to consider this approach as the winner as it obtains the best results in validation keeping a reasonable size for the model.

### 4.3. Test Results between Hadronic Models

As the SVR obtains on average the best results with validation data, this section tests the behaviour of this model with the test data sets. Firstly, the test data is the one obtained using the same hadronic model that provided the training/validation dataset. Afterwards, a comparison with data generated with the other model (EPOS LHC) will be done.

This is a very interesting experiment as it will show if the model was able to learn the natural phenomenon that both simulators are generating.

Figure 7 shows results for test data generated using QGSJET-II.

The metrics for the test data are shown in Table 6.

Figure 8a shows the results of the SVR when receiving as input the data generated by the EPOS LHC simulator.

With the following results shown in Table 7.

Both Figure 7 and Figure 8 represent the histogram of the error and a scatter plot of the real output versus the predicted output from the model for different types of particles: helium, iron, oxygen and proton for QGSJET-II, and iron and proton for EPOS LHC.

In both cases, for all particles, the histogram is centered in 0 and it has a narrow spread, indicating that the predictions are quite accurate. As Table 6 and Table 7, regardless of the simulator used and according to MAE and MSE metrics, particles such as protons seem more difficult to predict, probably due to the higher variability in the output value in comparison with the other particles.

When feeding the model with the EPOS LHC data, the behaviour is well reproduced although there are larger errors for particles that produce a higher number of muons in the shower. However, it is very interesting to observe that, as the model has learned from data from QGSJET-II, it tends to always underestimate the number of muons in comparison with the EPOS LHC output. This result confirms the study carried out in [39].

## 5. Conclusions

The use of machine learning models to provide solutions in the field of Physics is becoming popular. In this context, this paper has tackled the muon count estimation in water-Cherenkov tanks using simulations provided by the Pierre Auger Collaboration. The paper presents a comparative of the state-of-the-art models showing very good performance for all non-linear models regarding several error approximations. From the required space to implement the models, however, the results have shown that differences are critical and Neural Networks are the best ones. This aspect is usually ignored but should be carefully treated in monitoring and embedded systems. Finally, variable selection still is a must in the machine learning pipeline showing how the errors and model size are reduced when is applied. Overall, it is possible to conclude that the approaches presented in the paper are suitable to be deployed as a supplement to the improvement planned for the Observatory, in order to complement those stations not covered by Auger Prime, and to help monitoring the output provided by the physical systems in order to diagnose malfunction. Further experiments and analysis should be done using real data registered by the stations to analyse the homogeneity in the model’s behaviour.

## Figures and Tables

**Figure 1 entropy-22-01216-f001:**
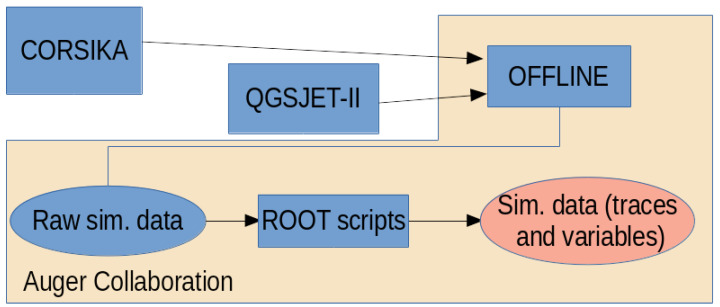
Data generation flow from Monte Carlo simulations until data are ready to be used by Machine Learning models.

**Figure 2 entropy-22-01216-f002:**
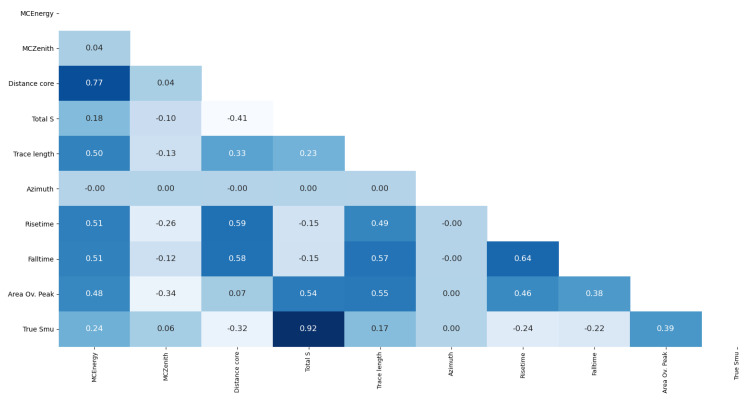
Correlation coefficients for each pair of variables.

**Figure 3 entropy-22-01216-f003:**
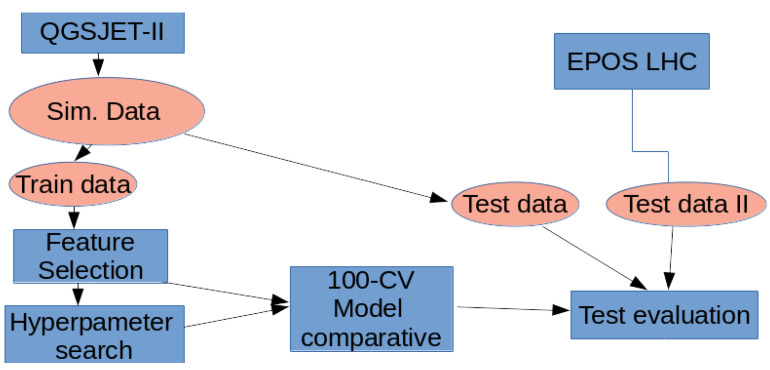
Experiments’ work flow of the comparative.

**Figure 4 entropy-22-01216-f004:**
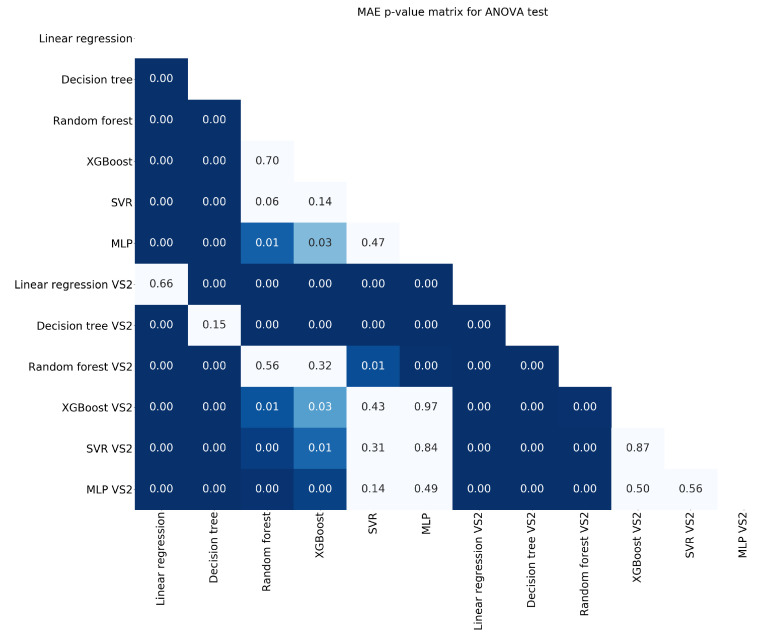
*p*-values for the ANOVA test using the criterion MAE.

**Figure 5 entropy-22-01216-f005:**
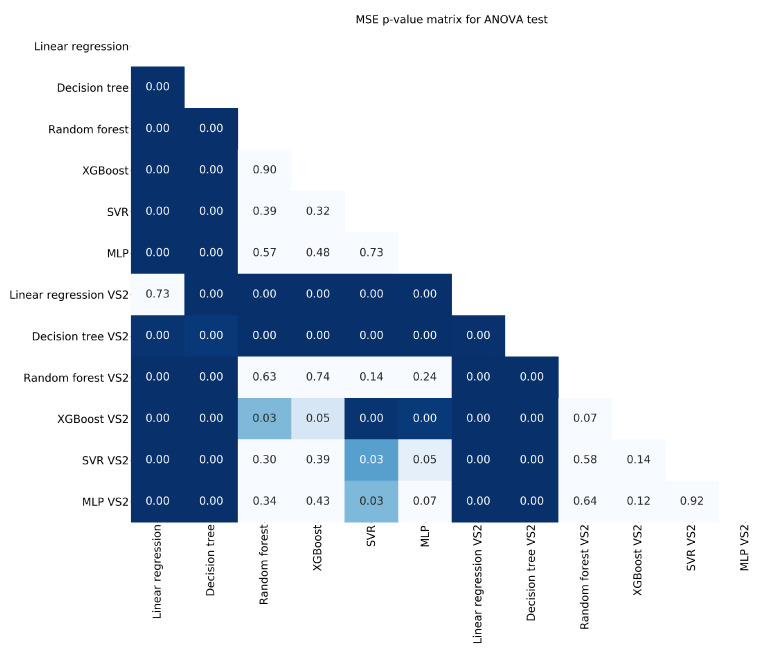
*p*-values for the ANOVA test using the criterion MSE.

**Figure 6 entropy-22-01216-f006:**
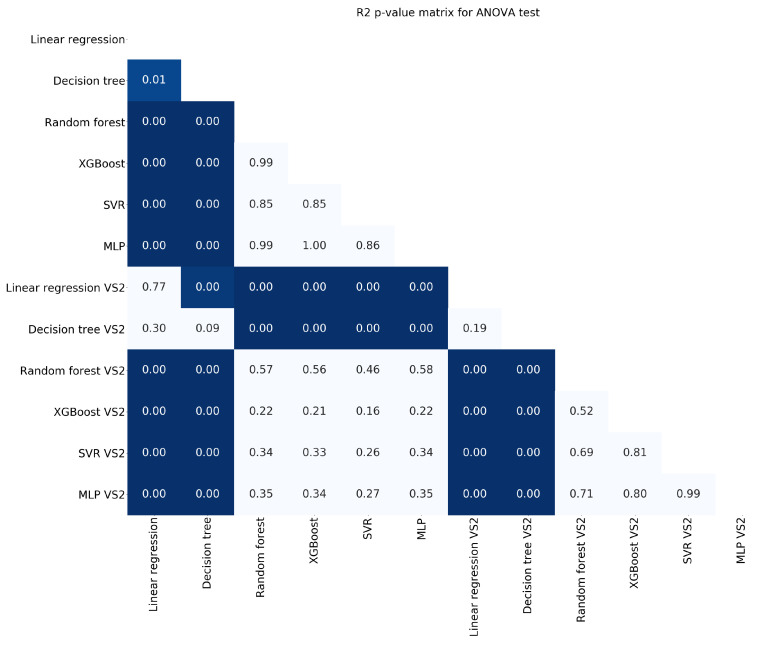
*p*-values for the ANOVA test using the criterion $R^{2}$.

**Figure 7 entropy-22-01216-f007:**
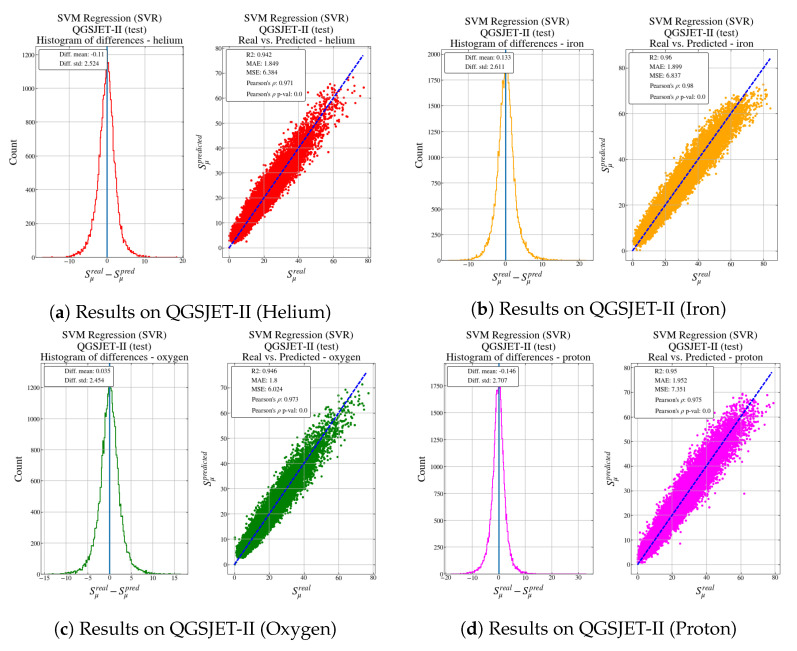
Results for the test data set generated using the QGSJET-II hadronic model.

**Figure 8 entropy-22-01216-f008:**
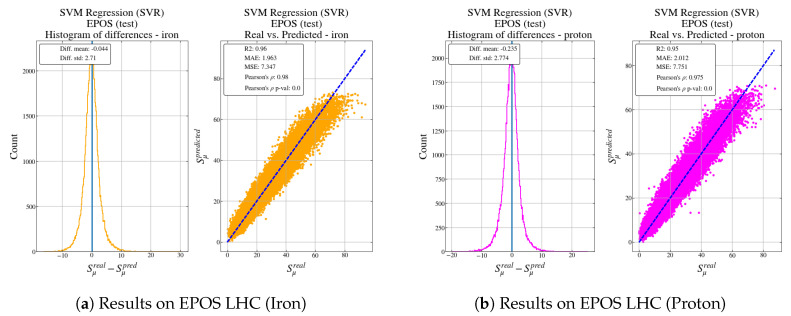
Results for test data set generated using the EPOS hadronic model.

**Table 1 entropy-22-01216-t001:** Ranking and scores obtained after applying the mRMR algorithm to perform variable selection.

Ranking	Score	Variable Name
1	0.2549	Total signal registered by the station
2	0.0368	Signal Risetime
3	0.0233	Signal Falltime
4	0.0189	Monte Carlo Energy
5	0.0183	Distance to the core
6	0.0113	Area over Peak
7	0.0070	Trace length
8	0.0067	Monte Carlo Zenith angle
9	0.0039	Azimuthal angle

**Table 2 entropy-22-01216-t002:** Feature importance as XGBoost criterion.

Ranking	Importance	Variable Name
1	234	Total signal registered by the station
2	94	Distance to the core
3	86	Signal Risetime
4	68	Monte Carlo Energy
5	62	Area over Peak
6	47	Monte Carlo Zenith angle
7	29	Signal Falltime
8	24	Azimuthal angle
9	2	Trace length

**Table 3 entropy-22-01216-t003:** Results for the mean and standard deviation (in brackets) for 100-fold CV using the variable selection XGBoost based.

	MAE	MSE	$R^{2}$
**LR**	2.53 (0.32)	12.09 (3.52)	0.87 (0.06)
**DT**	2.23 (0.20)	9.80 (2.17)	0.89 (0.05)
**RF**	1.82 (0.17)	6.62 (1.35)	0.92 (0.03)
**XGBoost**	1.83 (0.16)	6.65 (1.39)	0.92 (0.03)
**SVR**	1.86 (0.13)	6.47 (1.13)	0.93 (0.04)
**MLP**	1.87 (0.13)	6.52 (1.05)	0.92 (0.04)

**Table 4 entropy-22-01216-t004:** Results for the mean and standard deviation (in brackets) for 100-fold CV using the variable selection using mRMR.

	MAE	MSE	$R^{2}$
**LR**	2.55 (0.33)	12.27 (3.53)	0.86 (0.06)
**DT**	2.27 (0.18)	10.74 (2.07)	0.88 (0.06)
**RF**	1.81 (0.14)	6.71 (1.17)	0.92 (0.04)
**XGBoost**	1.87 (0.12)	7 (1.11)	0.92 (0.04)
**SVR**	1.88 (0.09)	6.79 (0.87)	0.92 (0.04)
**MLP**	1.89 (0.10)	6.78 (0.88)	0.92 (0.04)

**Table 5 entropy-22-01216-t005:** Resources required to store and access the models for the two subsets of features selected. Sizes are expressed in Mebibytes ($2^{20}\approx 10^{6}$ Megabyte) and Kibibytes ($2^{10}\approx 10^{3}$ Kilobyte).

	Space in Storage
**XGBoost**	646.7 MiB
**SVR**	4.2 MiB
**MLP**	183.6 KiB
**XGBoost-VS2**	29.4 MiB
**SVR-VS2**	4.2 MiB
**MLP-VS2**	74.1 KiB

**Table 6 entropy-22-01216-t006:** Metrics obtained for each type of particle.

Particle	MAE	MSE	$R^{2}$
Helium	1.849	6.384	0.94
Iron	1.899	6.837	0.96
Oxygen	1.8	6.024	0.94
Proton	1.952	7.351	0.95

**Table 7 entropy-22-01216-t007:** Metrics obtained for each type of particle using EPOS LHC simulations.

Particle	MAE	MSE	$R^{2}$
Iron	1.963	7.347	0.96
Proton	2.012	7.751	0.95

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
