# Peer review of "A Comparative Analysis of Machine Learning Techniques for Muon Count in UHECR Extensive Air-Showers"

_entropy, 2020, doi:10.3390/e22111216_

Round 1
Reviewer 1 Report
The article is easy to read and the introduction presents the question that seems to be addressed.
A first question appears in view of the data. They are simulated, that's a possibility. The question is more how the (expected) experimental imprecision was correctly integrated. This is an essential point and which seems difficult to me to understand here. we must therefore be much more critical on the issue.
In the same vein, no abbreviation has apparently been brought into test sets. This is a shame and it would be good to know how the approaches react to data outside the learned intervals.
With over 20,000 records, I would have expected 5-crossvalidation. It seems that is not the case. Why?
One of the problems with artificial data is that the description space is more harmonious than real experimental data. I think a cross validation would give very low standard deviations.
Was the 100-fold cross validation performed on the training sample, but only on the XGboost? The text is a bit confusing.
The results are not detailed enough, we need more criticism (Figures, Tables and little text). LR seems in fact sufficient, isn’t it?
The final question is the real usability of the approach apart from any theoretical data, it must be defended.
Author Response
Thank you for your valuable comments that have improved the quality of the paper.We have addressed all of them and modified the paper accordingly.
Please note that reviewers' comments are in bold while our answers are not. Additions to the
original manuscript are indicated in blue.

Reviewer 2 Report
A comparative analysis of machine learning techniques for muon count in UHECR extensive air-showers
By Alberto Guillén et al.
This paper compares six machine learning techniques to compute muon count by particle detectors, separating electromagnetic and muonic signals, and using two hadronic models for simulating data.
The paper is interesting and relevant to current high-energy primary cosmic rays research. However, the article badly needs refurbishment before been accepted. Consider also a grammar revision. Please find below a list of issues following the same order as the manuscript.
- Section 2: 'For the training, however, another testing stage will be carried out using the data available from simulations using the EPOS LHC model.'
- I understand that the training uses only the QGSJET-II package. The EPOS LHC model was only used to add additional testing data to discard dependencies on the simulation data source. Then, 'For the training, however…' is misleading.
- Section 2.1.: 'As stated, it seems straightforward as a blind source separation problem which can be tackled using Independent Component Analysis. However, the results obtained by our group using this method were far from being successful.'
- If using Independent Component Analysis seems straightforward, but it does not work, it is worth providing further explanations.
- Section 2.1.1: 'Nevertheless, according to the experts, the trace can be fully characterized by adding some features that can help the model to learn the function.'
- The authors probably refer to previously cited references, but they should be more specific than mentioning experts.
- The lack of proper references also applies to Section 4.1: 'As there are new variables proposed by the experts…'
- Section 3.
- I understand that an extended explanation of the different machine learning techniques is beyond this paper's scope. Still, it is not admissible to reduce the experiment descriptions to the programmer's logbook notes (especially in 3.4, 3.5, and 3.6).
- Have the authors developed the software for the models on their own? If so, they should state that in the text; otherwise, provide relevant information (e.g., libraries used, packages, APIs). Some open code developers may require proper acknowledgment.
- Section 3.5:
- 'This tool is an implementation [14] of the paradigm known as Gradient tree boosting which is also known as gradient boosting machine (GBM) or gradient boosted regression tree (GBRT), first presented in [15]'
- This sentence is quite wordy.
- eXtreme Gradient Boost (XGBoost)
- It is confusing that the authors use XGBoost as a model for comparing results (included in section 3 dedicated to models, compared with other models in Figures 4-6) and as a feature selection criterion (along with mRMR). Is it possible that using XGBoost as a selection criterium has any effect on the XGBoost model performance?
- 'This tool is an implementation [14] of the paradigm known as Gradient tree boosting which is also known as gradient boosting machine (GBM) or gradient boosted regression tree (GBRT), first presented in [15]'
- Section 3.6: 'Learning rate: Adpative using \alpha \in 0.0001, 0.001, 0.01'
- Include curly bracket delimiters: Learning rate: Adaptive using \alpha \in \{ 0.0001, 0.001, 0.01 \}
- Section 4.1: 'As Figure 2 shows, the maximum correlation between regressors is of 0.77 (energy and distance to the core).'
- However, the maximum correlation is 0.92 (Total S vs. True Smu).
- Figure 2 contains three variables not explained in the text: EventID, SimID, and Run Number. These variables' names seem to be related to the simulation procedure rather than measurements, and they should have no relevance (please, erase them).
- Refer to Section 2.1.1 comments above about the substitution of 'by the experts' by proper references.
- Section 4.1.1.
- I miss brief explanations about Mutual information and minimum redundancy and maximum relevance approach.
- 'Table 1 shows the feature ranking using the mRMR algorithm (Higher is Better).' I think that the authors mean that: the higher, the better, refers to the score rather than the ranking.
- Section 4.1.2 (pp 6 & 7): 'After comparing the results, it seems reasonable to keep the first six variables in both methods and compare them…'
- The authors take the six first ranked variables of the XGBoost method, but they are not the same as the six first ranked variables of the mRMR algorithm.
- The bullet list of variables does not correspond precisely to the names listed in Tables 1 and 2: 'Monte Carlo Zenith \theta' in the bullet list, but 'Zenith angle' in the tables.
- Section 4.1.2 (p 8).
- 'In order to determine which model/algorithm performed the best, a test ANOVA [21] was carried out to see if there were significant differences in the results.'
- Explain the ANOVA procedure used: how do you build the groups? What dependent and independent variables do you use? Figures 4-6 are not enough to explain the method used. Do the authors perform a One Way ANOVA for each cross-validation metric as the continuous dependent variable and the fitting models as categorical variables? Colors in figures 4-6 are related to p-values, but they should be explained.
- ANOVA requires sample independence and homogeneity of variances among the groups. Is the dependent, continuous variable obtained from the different fitting models, calculated from the same input data? In that case, the procedure violates the sample independence condition, and the results probably show a high degree of correlation. Besides, the results from different fitting models do not necessarily share the same variance.
- Table 5: KiB and MiB are computer units seldom used in Astronomy. Consider either to define KiB and MiB or to change to the most common KB and MB units.
- 'In order to determine which model/algorithm performed the best, a test ANOVA [21] was carried out to see if there were significant differences in the results.'
- Section 4.2.2: 'Even a deep MLP like the one selected for the first subset of variables fits in a few KibiBytes.'
- From the text and Table 5, the reader can deduce that the authors' arguments favor the MLP model as the most economical in space storage. But only in this last sentence they explicitly mention the MLP method. However, including 'Even a deep MLP…' they imply that other models perform even better than MLP.
- Section 4.3:
- Metric lists
- The metrics for the test data should be in tables rather than in lists.
- 'The scatter plots show that particles such as protons are more difficult to predict, due to the higher variability in the output value in comparison with the other particles.'
- This sentence is ambiguous. Which 'particles such as protons' do the authors mean. Only protons are elementary particles considered in this paper. Do they mean atomic nuclei, ions? Neither the scatter plots nor the metric lists provide evidence that the protons' scattering is significantly larger than the iron. The authors could perform a test for differences in the variances to sustain their assertion, but:
- Metric lists
-
-
-
- With such a large quantity of data points in their simulations, any tiny difference may be statistically, but not practically, significant.
- Also, the large quantity of simulated data may produce spurious effects that are not essential to the particles' nature but due to computation simplifications or incomplete understanding of the physics involved.
-
-
Author Response

(The authors gave the same response as above.)

Round 2
Reviewer 1 Report
The authors answered most of my questions.
The manuscript has been greatly improved.
Reviewer 2 Report
Thank you for attending to my concerns and clarify some issues.